# Fluorescence Image-Guided Navigation Surgery Using Indocyanine Green for Hepatoblastoma

**DOI:** 10.3390/children8111015

**Published:** 2021-11-05

**Authors:** Eiso Hiyama

**Affiliations:** 1Department of Pediatric Surgery, Hiroshima University Hospital, Hiroshima 734-8551, Japan; eiso@hiroshima-u.ac.jp; Tel.: +81-82-257-5951; 2Natural Science Center for Basic Research and Development, Hiroshima University, Hiroshima 734-8551, Japan

**Keywords:** indocyanine green, navigation, hepatoblastoma, hepatectomy, metastasectomy, tumor burden, resectability, outcome

## Abstract

In the past decade, navigation surgery using fluorescent indocyanine green (ICG) dye for hepatoblastoma (HB) has been developed for the resection of primary or metastatic tumors. Since HB cells can take up ICG but cannot excrete it to the bile duct, ICG remains in the HB cells, which can be used for navigation by fluorescent activation. The complete resection of the primary tumor as well as metastatic tumors, along with appropriate neoadjuvant and adjuvant chemotherapy, is essential for cure. ICG fluorescence can detect microscopic residual lesions in the primary lesion and identify micro-metastases in the lung or other lesions; consequently, ICG navigation surgery may improve outcomes for patients with HB. The basic technique and recent advances in ICG navigation for HB surgery are reviewed.

## 1. Introduction

Hepatoblastoma (HB) is the most common pediatric malignant liver tumor, and is usually diagnosed in children under three years of age. HBs are classified according to international risk group by the Children’s Hepatic tumors International Consortium (CHIC) [1]. Since the outcomes for patients with HB depend on the complete resection of the tumor, patients with low-risk HBs, which are usually resectable at diagnosis, have more than 85% survival. In patients with intermediate-risk tumors, which are unresectable at diagnosis due to PRETEXT IV and/or positive annotation factors, such as portal vein invasion, hepatic vein invasion, and tumor rupture, the outcomes of those whose tumors become resectable with neoadjuvant chemotherapy are favorable [2,3]. In patients with high-risk tumors, who usually have lung metastases, those whose metastases are diminished by chemotherapy or completely resected by thoracotomy have favorable outcomes [4]. Therefore, the complete resection of the primary liver tumor and the diminishment of metastases by chemotherapy and/or thoracotomy may be essential for the cure of patients with HB. Since HB primary tumors are usually diagnosed as large hepatic tumors, they are difficult to resect with a sufficient surgical safety margin. Therefore, more precise resection using navigation for the existence of tumor cells might be effective for improving outcomes for patients with HB.

ICG is an organic anion that is almost exclusively taken up by the liver and rapidly excreted into the bile without undergoing biotransformation or enterohepatic circulation. Therefore, ICG clearance has been used as a valuable tool for identifying patients with impaired liver function and, in liver surgery, identifying patients at risk of developing postoperative liver dysfunction or surgical complications [5,6]. ICG is a safe and convenient tool for liver surgeons [7,8]. ICG is also taken up by malignant liver tumor cells, such as hepatocellular carcinoma, hepatoblastoma, and others. However, malignant cells do not excrete ICG because they do not have a connection to the bile duct. Therefore, ICG remains only in malignant hepatic cells after clearance from normal hepatic cells. Since ICG is effectively excited under far-infrared ray (FIR), ICG is useful for navigation to detect malignant hepatic cells.

In this review, which focuses on ICG fluorescence-guided navigation surgery for HB, we discuss the development, underlying ICG uptake mechanism, clinical applications, and future potential of this technology.

## 2. Methodologies for ICG Navigation Surgery in HB

When ICG is administered intravenously, normal hepatocytes take it up uniformly over 5–10 min and retain it for approximately 24 h [9,10]. Hepatocytes in a cirrhotic liver retain ICG longer than normal hepatocytes. In ICG navigation studies for hepatocellular carcinoma (HCC), ICG (0.5 mg/kg) was administered intravenously between 1 and 14 days [6,11]. In some reports, no visualization was reported when ICG was administered three weeks or more before surgery [12]. On the other hand, ICG remains in the bile duct and intestine for 2 days after injection, sometimes resulting in false positive signals in abdominal surgery. Therefore, the administration of ICG is recommended at 3 days or more before abdominal surgery, but one day before thoracotomy.

### 2.1. Appropriate Protocol of ICG-Navigated Surgery

In the original ICG navigation study for HCC, ICG (0.5 mg/kg) was administered two weeks prior to surgery [13]. In previous studies, ICG injection times of one day to several weeks before surgery were investigated in patients with HCC [9,10], but a few instances of non-visualization were reported when ICG was administered too early before surgery [12]. Therefore, ICG should be administered within two weeks before surgery.

Just after the intraoperative administration of ICG, intrahepatic tumors are identified as a shadow on the Photodynamic Eye (PDE) system. They are noticed as non-fluorescing negative lesions because ICG is visualized as contrast media in vessels and capillary vessels [14,15]. Next, ICG is imported into hepatocytes and excreted into the small bile duct. As a well-known ICG test, the uptake of ICG into hepatocytes is sometimes delayed in liver diseases, including liver cirrhosis, fibrosis and others. Therefore, most reports mention that a two-day interval might be more effective for patients with advanced cirrhosis, since the signal intensity of the noncancerous liver parenchyma may interfere with detection [13].

To determine the optimal dose of ICG for navigation surgery, investigators assessed a range of doses from 5 mg/body to 20 mg/body or from 0.25 mg/kg to 0.5 mg/kg [6]. It was revealed that smaller doses (1.25 mg to 2.5 mg/body for adult patients) were optimal for navigating liver tumors, providing sufficient signals to achieve the objective [16,17]. On the other hand, ICG injection more than one day before surgery might increase false-positive nodules in cirrhosis or post-chemotherapeutic liver in adult experiences [6]. Therefore, the injection dose and timing should be chosen according to the status of patient’s liver.

### 2.2. Intraoperative Procedure for the Detection of ICG Positive Lesions

After ICG is administered to the patient, navigation using ICG requires exposure to excitation light containing 760 nm infrared rays and collection of emitted fluorescence as 830–870 nm (Figure 1).

Several commercial detectors for near-infrared fluorescence devices are classified into two types: handle and endoscopic types [8]. The handle type, such as the PDE system^®^, is applied in open surgery, and the endoscopic type is used for laparoscopic or thoracoscopic surgery. When the PDE system is used for open surgery for HCC and HB, the operating light should be turned off to detect the near infrared light by ICG, but the operating light does not interfere with the endoscopic type. The other detector for ICG navigation is an endoscopic system that has been used in laparoscopic and thoracoscopic surgery. These systems, such the Olympus: VISERA ELITE System^®^ and the Karl Stolz: D-Light P System^®^, have the working capacity to switch between white and infrared-red light with a push bottom. The PINPOINT system allows for a simultaneous overlay view of the normal white-light mode and the infrared-red mode with the same range through a single endoscope, allowing surgeons to perform surgery in real time without switching.

## 3. Current Status of ICG Navigation Surgery for HB

### 3.1. Previous Reports on ICG Navigation Surgery for HB

The PDE system has been used for open surgery for primary and residual tumors in HB (Figure 2). Most cases were reported by Japanese colleagues [16,17,18,19,20]. In those studies, ICG (0.5 mg/kg) was administered intravenously 60–138 h prior to surgery (Table 1). In all cases, specific fluorescence visualization of hepatic lesions was detected in the liver; therefore, the benefit of this procedure is to evaluate negative margins after tumor resection. Since HB tumors are sometimes very large, making them difficult to resect with a sufficient negative safety margin, the evaluation of marginal negativity is important. Moreover, for multifocal HB tumors, ICG navigation was useful for the detection of small separate lesions in the liver. In fact, small tumors with 8–10 mm diameter were detectable by ICG navigation. Since ICG is excreted into the biliary system, followed by the bowel loops, fluorescence in the bowel loops may also produce pseudo-positive lesions. Therefore, ICG should be administered more than 72 h prior to guided surgery in order to minimize non-specific fluorescence signals in the bile duct and bowel loops. However, bile stasis sometimes occurs near the margin of the tumor so that ICG remains near the tumor stasis.

### 3.2. Previous Reports of ICG Navigation Surgery for Primary Tumors in HB

ICG navigation surgery has been performed for primary liver tumors in more than 20 cases of HB (Table 1). Hemihepatectomy or extended right hepatectomy was performed for these patients, and liver transplantation was reported for two patients. ICG (0.5 mg/kg) was administered 72–96 h prior to the primary hepatectomy in a case reported by Yamamichi et al. [20], 60–138 h prior to surgery in the cases reported by Souzaki (*n* = 4) [16], and 72 h prior to surgery in the cases reported by Yamada (*n* = 12). The ICG-navigated lesions of primary HB tumors were visualized in all cases, suggesting that approximately 72 h before surgery is an appropriate time for ICG injection. ICG navigation for liver surgery was useful for detecting the residual tumor at the resection margin, such as surrounding the inferior vena cava and invasion into the diaphragm or extended lobe and small tumor nodules in multifocal HB (Figure 3). ICG is excreted into the biliary system from hepatocytes and then into the intestine, so fluorescence in the bile duct and intestine may occur as a non-specific fluorescence. HB occasionally develops lymph node metastasis and dissemination into the abdomen. These extrahepatic lesions become detectable at 72 h after ICG injection with minimal background fluorescence in the intestine. When a large tumor blocks bile excretion or intestinal passage, the ICG remains in the bile duct or intestine near the tumor, which might be detected as a false positive signal.

Based on these findings, the recommended period of ICG injection (0.5 mg/kg) is approximately 72 h before abdominal surgery for HB.

### 3.3. Previous Reports on ICG Navigation Surgery for Lung Metastasis in HB

The outcome for patients with HB with lung metastases is well-known to depend on the diminishment of metastatic lesions by chemotherapy and surgical resection. Event-free survival in these cases depends on margin-negative resection. Moreover, liver transplantation for advanced HB is permitted for cases whose lung metastases have been completely cleared. Therefore, more accurate detection of micro-metastasis and microscopic residual tumors is thought to contribute to improving the outcome for these patients. In thoracic surgery to resect pulmonary metastases, there is no interference from residual fluorescence in the liver, bile duct or bowels, and ICG is clearly detected after administration only 24 h before surgery. Therefore, in patients with HB, ICG navigation surgery was initially developed for lung metastasectomy [21,23]. In navigation surgery in the abdomen, ICG remains in non-cancerous tissues such as the liver, bile duct and bowel at 24 h after injection. Therefore, ICG should be administered more than 72 h before surgery.

In representative intraoperative visualizations (Table 1), 250 fluorescence-positive lesions were successfully identified in 37 surgeries on 10 patients. Among them, the smallest detectable nodule was 0.062 mm. Souzaki et al. performed 6 pulmonary metastectomies with ICG navigation and found metastases as small as 1.2 mm [16]. As shown in Table 1, in our one false-negative lesion (for which the reason this metastatic lesion failed to be visualized was unclear), as the same patient later underwent another ICG-guided surgery, this metastasis was then visualized, but the tumor was located more than 10 mm from the lung surface. Pushing this tumor to the lung surface led to obtaining a clear fluorescent signal in this lesion, indicating a limitation on the detectable depth of ICG signals (Figure 4).

On the other hand, there are several false positive signals, which are usually weak. Histologically, ICG signals are detectable using fluorescence microscopy covering an infrared light field. In our experience, histological examination revealed that false positive nodules include sticky positive cells that are inflammatory, such as macrophages or lymphocytes. Therefore, ICG might sometimes be taken up into inflammatory cells that demonstrate positive ICG signals (Figure 5). After ICG injection, ICG binds to albumin and lipoprotein and is taken up by endocytosis in hepatocyte and inflammatory cells and then excreted rapidly [24]. The mechanism of the ICG excretion defect is obscure. In reported sentinel node navigation surgeries using ICG, false positive signals were sometimes detected in non-sentinel nodes, suggesting that lymphocytes sometimes retain ICG [25]. In addition, the bile duct and intestine showed false positive signals from the remaining ICG excreted by hepatocytes.

### 3.4. Previous Reports on ICG Navigation Surgery for Other Lesions in HB

For the detection of lymph node metastasis, ICG navigation has the sensitivity to distinguish metastatic nodes from normal ones. ICG was also detectable in abdominal wall invasion and omental metastasis. Tumors in the pancreas and bone were reported as ICG-negative [23]. These lesions might exist in areas too deep to be detected by infra-red rays.

The number of such cases with extrahepatic lesions is so small that more experience of detecting ICG is required for the evaluation of ICG navigation in surgery for extrahepatic lesions. For the resection of metastatic lymph nodes in HB, ICG navigation was also effective, especially in the mediastinal or abdominal cavities. In the abdominal cavity, ICG is excreted into the bowels, which may interfere with the detection of positive lesions; thus, ICG should be administered around 72–96 h prior to surgery. In cases with metastases to the pleura and diaphragm, ICG navigation surgery has greatly helped surgeons to identify the exact location and extent of the tumor due to the passage of ICG through abnormal permeable tumoral vessels.

## 4. Conclusions

ICG fluorescence navigation surgery can be used safely and easily to identify the primary tumor and metastatic hepatoblastoma in real time during open and laparoscopic pneumonectomy and hepatobiliary surgery. With further developments in cancer-specific fluorophores and imaging systems, intraoperative fluorescence imaging will develop into an essential navigation tool and a standard surgical procedure for evaluating tumor cell spread, micrometastasis, and the risk of postoperative recurrence.

## Figures and Tables

**Figure 1 children-08-01015-f001:**
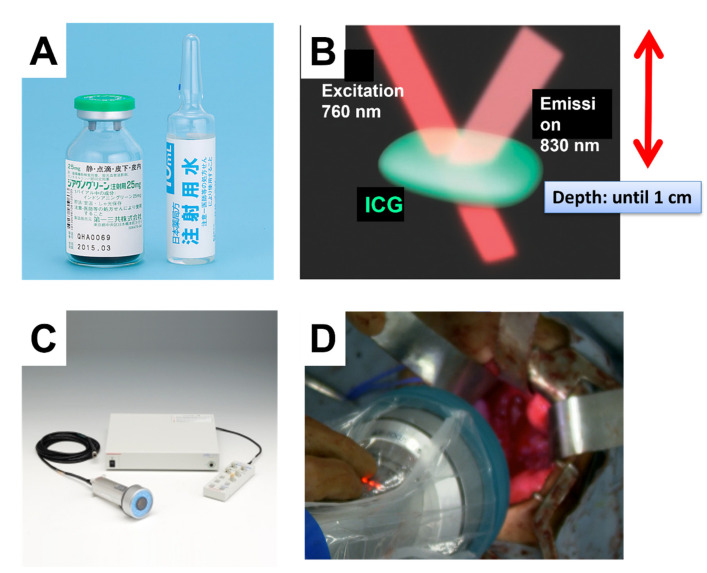
PDE system customized for the detection of ICG fluorescence (Hamamatsu Photonics: PDE System^®^). (**A**) Indocyanine green (ICG) vial. (**B**) Schema of ICG detection. The excitation light is 760 nm, and the emission signal is 830 nm. (**C**) PDF detector (handle type). (**D**) Intraoperative detection of ICG signals.

**Figure 2 children-08-01015-f002:**
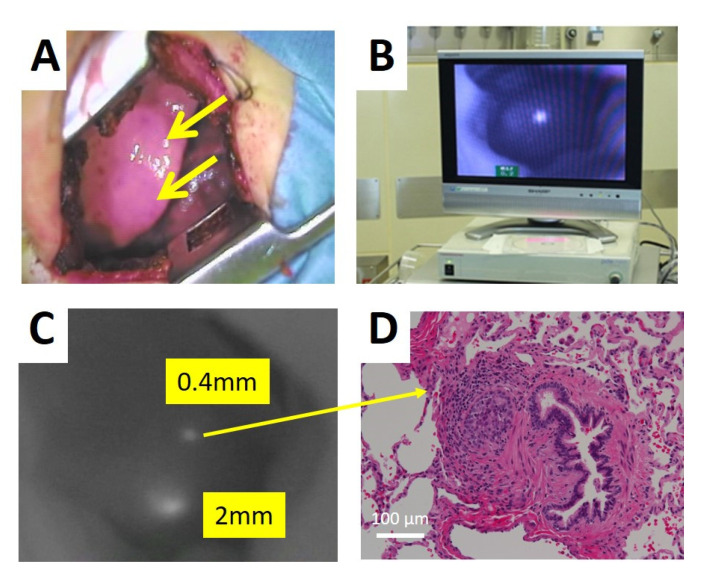
PDE system for the detection of lung metastases by ICG fluorescence (Hamamatsu Photonics: PDE System^®^). (**A**) Surgical field under thoracotomy. The metastatic nodules were invisible. (**B**) Video screen of ICG navigation surgery. (**C**) ICG positive nodules. (**D**) The pathological finding of a small ICG positive nodule.

**Figure 3 children-08-01015-f003:**
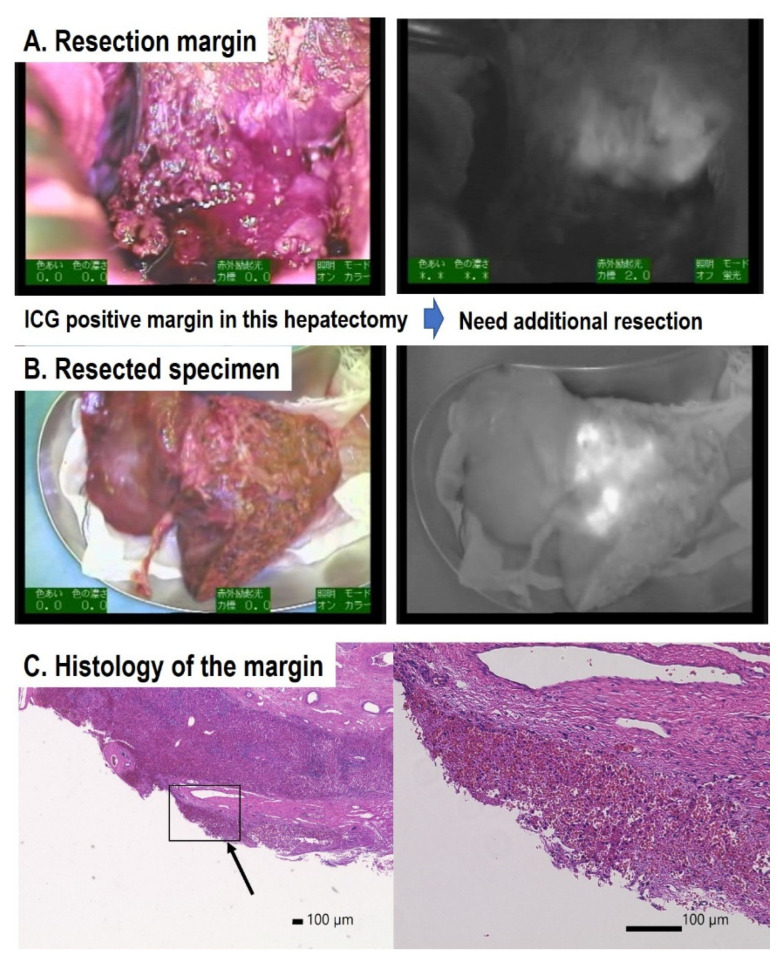
Detection of marginal residual tumor cells by ICG fluorescence. (**A**) Resected margin: ICG was positive. (**B**) Resected specimen: resected margin was also ICG positive. (**C**) Microscopic examination of the margin. Histological examination suggests that the margin of the resected specimen might be positive (arrow). The higher-magnification field revealed the existence of malignant tumor cells.

**Figure 4 children-08-01015-f004:**
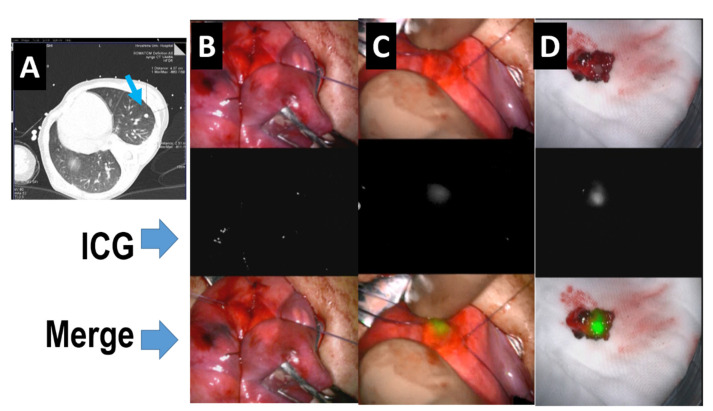
ICG detection of a deep lung metastasis by ICG fluorescence. (**A**) CT scan shows the existence of a lung metastasis. (**B**) Video screenshot of ICG navigation surgery. (**C**) ICG positive nodules. (**D**) The pathological finding of a small ICG-positive nodule.

**Figure 5 children-08-01015-f005:**
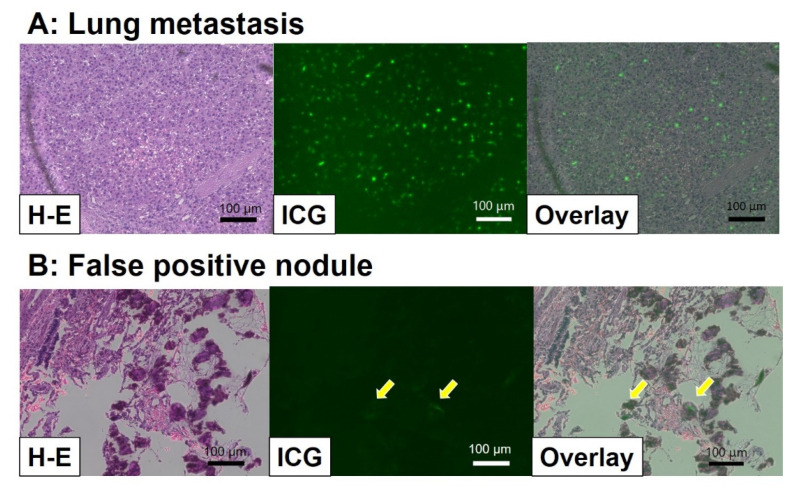
Histological examination via infrared fluorescence microscopy. (**A**) Lung metastasis: strong ICG signals were detectable at the tumor cells. (**B**) False positive nodule: in this lesion, weak and sticky positive signals were detected, as indicated by asterisks. These cells were considered to be macrophages.

**Table 1 children-08-01015-t001:** Reported ICG navigation surgery in hepatoblastoma.

Reports	Surgery/Cases	Location (*n*: Surgery)	ICG Injection Time before Surgery (hrs)	ICG Positive Lesions	Size (mm) (Diameter)	Histology
Kitagawa N. et al. [21]	37/10	lung (37)	24	250	minimum-0.062	29 false positive, 5 negative nodules (pathological negative)
Yamamichi, T. et al. [20]	3/3	liver (2)	72–96	2	11, 70	
		lung (1)	72–96	24	>3	
Toyofumi F. et al. [18]	1/1	lung (1)	24	1	unknown	
Souzaki, R. et al. [16]	10/5	liver (4)	60–180	4	40–130	
		lung (4)	18–27	11	1.2–15	1 false positive
Yamada, Y. et al. [17]	36/20	liver (13)	72	13	8–130	1 false positive at margin
		lung (17)	24	30	1–12	6 false positive, 1 false negative
		others (6)	72–96	4	16–31	2 negative (pancreas, bone metastasis) 4 positive (1 lymph node, 2 peritoneal, 1 pleural and diaphragm)
Hiyama et al. [22]	19/12	liver (4)	72	5	10–150	2 false positive at margin
		lung (11)	24	54	1–28	1 false negative (positive at later surgery)
		others (3)	24	5	3–25	4 lymph nodes and 1 peritoneal metastasis

ICG: indocyanine green. The injection dose of ICG was 0.5 mg/kg in all reports.

## Data Availability

Since this is a review article, no new data were created or analyzed in this study. Data sharing is not applicable to this article.

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
