# Peer review of "Fluorescence Image-Guided Navigation Surgery Using Indocyanine Green for Hepatoblastoma"

_children, 2021, doi:10.3390/children8111015_

Round 1

Reviewer 1 Report

Dear author, it is a good brief of this issue. You should resend better quality of all pictures above all figure 1, figure 4 and figure 5. In the other hand, in the Table 1, if the ICG dose in all studies was the same you should delete the column and describe in the text or in the legend of Table.

Author Response

Thank  you for your prompt review. According to your suggestion, we remade the Fig. 1 with high quality picture. And others will be changed as possible as we could. And we deleted the column of ICG dose from Table 1 and added the dose in it’s figure legend. (page 6).

Reviewer 2 Report

The article entitled "Fluorescence image-guided navigation surgery using indocya-2 nine green for hepatoblastoma" is an interesting review summarizing the potential use of ICG to analyze negative margins after tumor resection, as well as the identification of metastases. However, it presents the problem that ICG can give false positives when visualized in parts such as the bowel. Overall the article is well structured and easy to understand. 
The only thing I missed, since the author mentions that it can give false positives when marking macrophages or that healthy cells release it but malignant cells retain it, is the mention of some in vitro and in vivo studies analyzing this uptake/release in different cell types and tumors. 

Minor points: 

  • ICG and HCC abbreviations must be defined. 
  • In line 44 a space is missing between dysfunction and or. 

Author Response

Thank you for you prompt reply, we added the sentences for explanation of false possibility in page 7 (lines 189-194).

And we corrected these misspellings you indicated.

Thank you.